# Mutagenesis of the ADAM17-phosphatidylserine–binding motif leads to embryonic lethality in mice

Martin Veit, Björn Ahrens, Jana Seidel, Anselm Sommer, Sucharit Bhakdi, Karina Reiss

ADAM17, prominent member of the "Disintegrin and Metalloproteinase" (ADAM) family, controls vital cellular functions through cleavage of transmembrane substrates. Several of these play central roles in oncogenesis and inflammation, yet despite its importance, the mechanism by which ADAM17 is activated is not fully understood. We recently presented evidence that surface exposure of phosphatidylserine (PS) is the penultimate event required for sheddase activation, which occurs upon binding of a membrane-proximal, cationic binding motif to the anionic phospholipid headgroup. Here, we show that mutagenesis of the 3 amino acids constituting the PS-binding motif leads to embryonic lethality in mice. Heterozygotes showed no abnormalities. Primary hepatocytes and fibroblasts were analysed and found to express the mutant protease on the cell surface. However, PMA-stimulated release of ADAM17 substrates was completely abolished. The results directly support the novel concept of transiently externalised PS as essential trigger of extracellular protease function in vivo.

## Introduction

ADAM17, originally discovered as the TNF-$\alpha$ converting enzyme (1, 2, 3), has emerged as a pre-eminent member of the "a disintegrin and metalloprotease" family of transmembrane proteinases. More than 80 ADAM17 targets have been identified to date, prominent among which are cytokines, adhesion molecules, and cell surface receptors, including TNFR1 (4, 5, 6, 7). Moreover, ADAM17 regulates cell growth through the liberation of epidermal growth factor receptor (EGFR) ligands such as amphiregulin (AREG) or TGF-$\alpha$ (8, 9, 10). In humans, ADAM17 deficiency results in severe inflammatory skin and bowel disease, underlining its important role for epithelial cell homeostasis (11, 12). In vivo mouse models further emphasise the vital importance of ADAM17 (3, 13, 14, 15, 16, 17). Targeted deletion of exon 11 encoding the catalytic site of the protease ($Tace^{\Delta Zn/\Delta Zn}$) (3) resulted in the death of mice between embryonic day (E) 17 and

the first day after birth because of developmental defects in brain, heart, lung, skin, skeletal, and immune system (3, 18).

Activation of ADAM17 can be provoked by a heterogeneous spectrum of physiologic and non-physiologic stimuli. How the different activation pathways should converge to up-regulate sheddase function at the cell surface has remained an enigma. We recently presented evidence that transient breakdown of phospholipid asymmetry and exposure of the negatively charged phospholipid phosphatidylserine (PS) is key to the solution (17, 19, 20). A cationic PS-binding motif was identified in the membrane-proximal domain of ADAM17 that was required for activation of the sheddase in cultured cells (17). The question whether this motif was of relevance in vivo remained unanswered. In this investigation, mice were generated with a targeted deletion of the PS-binding motif comprising a small cluster of just three cationic amino acids. We found that this resulted in the death of mice between embryonic days E14–15. Although the mutated protease was still expressed on cells, the classical substrate TNFR1 could not be cleaved from the surface of primary murine hepatocytes or fibroblasts (MEFs). In addition, PMA-stimulated release of TGF-$\alpha$ and AREG was abrogated in the ADAM17$^{3x/3x}$ MEFs. Our results provide direct evidence that interaction of ADAM17 with externalised PS is the prerequisite for the protease to fulfil its sheddase function.

## Results

### Mutation of the PS-binding motif leads to early embryonic lethality

Asymmetrical distribution of lipids in the cell membrane is a hallmark of the living cell. The most important negatively charged phospholipids PS and phosphatidylinositol species reside predominantly in the cytoplasmic leaflet, where they serve as essential cofactors for many membrane-bound enzymes, including protein kinase C, phosphatase PTEN (Phosphatidylinositol 3,4,5-trisphosphate 3-phosphatase and dual-specificity protein phosphatase), tyrosine kinase c-Src, and MARCKS (Myristoylated alanine-rich C-kinase substrate) (21). Whereas the importance of PS–protein interaction is well established for intracellular proteins, the relevance of transient

Department of Dermatology, University of Kiel, Kiel, Germany

Correspondence: kreiss@dermatology.uni-kiel.de

phospholipid externalisation is unknown. The starting point of our investigation was the observation that sheddase function of ADAM17 apparently required interaction of the protease with surface-exposed PS ([17]). 3D-heteronuclear nuclear magnetic resonance experiments identified a small cluster of cationic amino acids R625/K626/G627/K628 as the putative PS-binding motif. Now, we provide in vivo evidence in support of the contention that surface-exposed PS plays a critical role in the control of ADAM17 sheddase function.

*ADAM17* was manipulated by CRISPR/Cas9 gene editing. The targeting strategy is depicted in Fig S1A. Mice were genotyped by PCR and digestion of the PCR product as shown in Fig S1B.

Heterozygous ADAM17$^{WT/3x}$ mice showed no abnormalities and were fertile. They were intercrossed to generate homozygous mice. No viable offsprings were produced. Consequently, ADAM17$^{3x/3x}$ embryos were analysed at different stages of embryonic development. Up to day 14 of embryogenesis, embryos presented in a Mendelian frequency as shown in Table 1. No viable ADAM17$^{3x/3x}$ embryos were obtained beyond E16. ADAM17$^{3x/3x}$ mice were smaller in size and about 50% presented with massive haemorrhages (Fig 1) as described for ADAM17 KO mice, which were likely due to impaired vessel formation. The penetrance of this phenotype was comparable with the classical KO which also showed haemorrhages in about 50% of the E14.5 animals ([18]).

## ADAM17 protein expression and cell surface localisation is not affected

It was first ensured that the protease and mutants showed a comparable expression in different tissues and cells. As shown in Fig 2A, both pro- and mature ADAM17 were detected in the liver and brain of the animals. MEFs and hepatocytes were derived from dissociated E14 embryos and maintained in culture. These cells also showed no difference in ADAM17 expression. Biotinylation experiments were performed to probe cell surface localisation, which was also found to be comparable with WT cells in both hepatocytes and MEFs (Fig 2B and C). The significant proportion of immature ADAM17 on the cell surface of hepatocytes (Fig 2B) was somewhat surprising with regard to the data published for other cells ([22], [23], [24]). However, cell surface localisation of pro-ADAM17 has been described ([25]) and, as known for pro-ADAM10 ([26], [27], [28]), probably varies depending on the transport machinery, the cell type, and the stage of embryonal development. The difference in relative amounts of surface-expressed protein in hepatocytes versus MEFs was possibly due to cell-specific

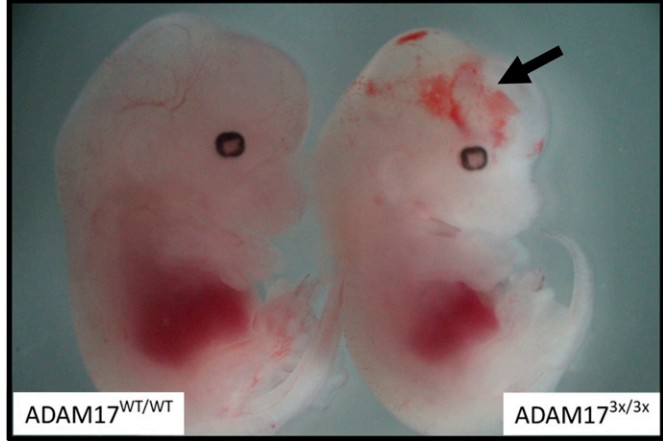

**Figure 1.  ADAM17$^{3x/3x}$ embryos develop internal haemorrhage.**
Whole-mount views of ADAM17$^{WT/WT}$ (left) and ADAM17$^{3x/3x}$ embryos (right) at E14. ADAM17$^{3x/3x}$ embryos exhibited haemorrhagic lesions compared with their WT littermates (see arrow).

variations in trafficking behaviour. Surface expression was sufficient to permit flow cytometric analysis in hepatocytes, and similar expression of WT and mutant ADAM17 protease was also found with this method (Fig S2).

## PMA-induced TNFR1 release is abolished in ADAM17$^{3x/3x}$ cells

The decisive question remained whether ADAM17$^{3x/3x}$ cells would indeed show impaired shedding capacity. In contrast to the closely related protease ADAM10, ADAM17 is well known to respond to PMA stimulation ([29]). Transient PS exposure was previously noted to occur in keratinocytes exposed to the stimulus ([17]). Hepatocytes of wild-type (WT) and ADAM17$^{3x/3x}$ animals also responded with comparable PS exposure upon PMA stimulation (Fig S3). Hepatocytes express substantive levels of TNFR1 whose cleavage by ADAM17 can readily be assessed ([4], [5], [30]). As shown in Fig 3, shedding of this substrate was significantly provoked by PMA in wild-type and heterozygous cells and abrogated in the presence of the metalloprotease inhibitor marimastat (MM). In striking contrast, constitutive shedding of TNFR1 appeared to be absent in ADAM17$^{3x/3x}$ hepatocytes, and no cleavage could be induced by treatment with the phorbol ester. A comparable picture was obtained for MEFs. Even though cells of both genotypes were PS positive (Fig S4A and B), the amount of soluble TNFR1 was significantly reduced and could not be increased upon PMA stimulation in the ADAM17$^{3x/3x}$ cells (Fig S4C). Interestingly, constitutive metalloprotease-dependent release of TNFR1 also appeared to be absent in these cells.

## Release of ADAM17 substrates is impaired in ADAM17$^{3x/3x}$ MEFs

To examine whether shedding of other ADAM17 substrates was similarly impaired, MEFs were transfected with AP-tagged AREG or TGF-α and stimulated with PMA. AP activity in the supernatant and cell lysates was determined as readout for substrate release (Fig 4A and B). Although PMA significantly increased ADAM17-mediated substrate release in WT MEFs, shedding could not be induced in ADAM17$^{3x/3x}$ cells. To ensure that PMA-induced shedding was

**Table 1.  Mutation of the PS-binding motif leads to early embryonic lethality.**

| C57BL/6N | Total | ADAM17$^{WT/WT}$ | ADAM17$^{WT/3x}$ | ADAM17$^{3x/3x}$ |
|---|---|---|---|---|
| Postnatal | 541 | 207 | 334 | 0 |
| ~E18 | 31 | 8 | 22 | (1) |
| ~E16 | 16 | 3 | 11 | (2) |
| ~E14 | 314 | 77 (4) | 148 (8) | 67 (10) |

Embryos were isolated at different stages of development. Up to day 14 of embryogenesis (E14) embryos presented in a Mendelian frequency. No viable ADAM17$^{3x/3x}$ embryos were obtained beyond E16. (X) = number of dead animals.

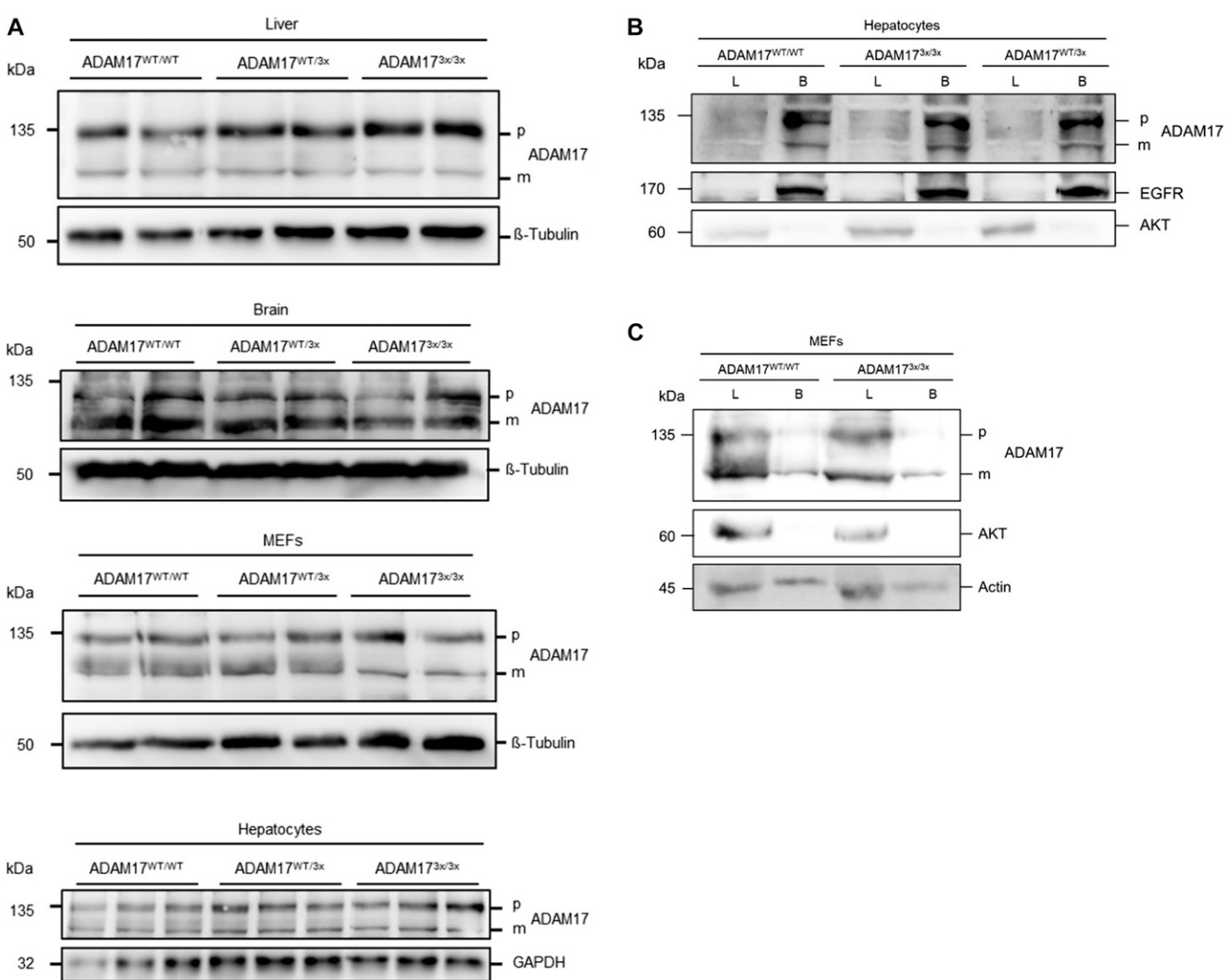

**Figure 2. Expression and surface localisation of ADAM17 is not affected.**
**(A)** Expression of ADAM17 pro- (p) and mature (m) form was analysed via immunoblot at E14 in the liver, brain, and in primary isolated fibroblasts (MEFs) and hepatocytes. No differences were observed between the different genotypes. **(B, C)** Biotinylation experiments confirmed cell surface localisation of mature ADAM17 in primary hepatocytes (B) and MEFs (C). Cells were biotinylated and whole lysates (L) were compared with biotinylated proteins (B). EGFR or actin detection was used as positive control for cell surface proteins and AKT was used as control to exclude the detection of cytosolic proteins.

indeed PS dependent, competition experiments were conducted with the soluble PS headgroup O-phospho-L-serine (OPS). As shown in Fig 4A and B, OPS dose-dependently decreased PMA-induced shedding.

It was particularly noteworthy that the constitutive release of both substrates was also significantly diminished in the ADAM17$^{3x/3x}$ MEFs. To further probe the relevance of PS binding for non-stimulated substrate release, we analysed the constitutive shedding of TGF-$\alpha$ over time in the presence or absence of OPS and MM. As shown in Fig 4C, substrate release increased over time. Upon short-term incubation of 30 min, OPS inhibition was similarly effective as MM. Dose-dependent inhibition by OPS continued to occur upon prolonged incubation, although the extent was diminished compared with MM.

## Discussion

After the first description of ADAM17-loss of function due to deletion of the catalytic centre (1), a number of further murine models have been established. Mice expressing hypomorphic *Adam17* alleles have been described where reduced ADAM17 levels lead to the development of eye, heart, and skin defects as a consequence of impaired EGFR signalling (16). ADAM17 activity is also controlled by iRhom1 and iRhom2, members of a family of evolutionarily related multi-pass membrane proteins (23, 31, 32). iRhom1/2-double deficient KO mice resemble ADAM17-deficient mice in that they die perinatally with open eyes, misshapen heart valves, and growth plate defects due to reduced EGFR phosphorylation (33). Our ADAM17$^{3x/3x}$ mice showed classical haemorrhages as described for

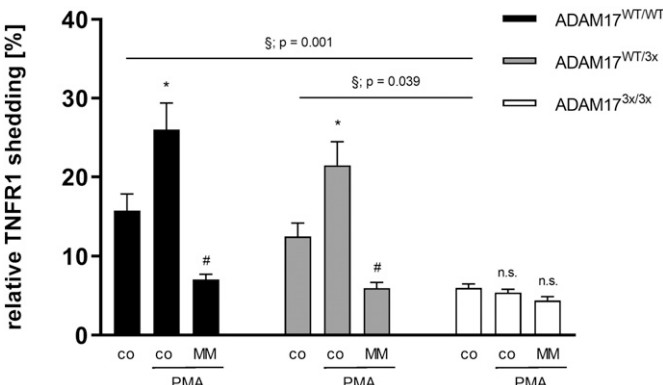

**Figure 3. Shedding of TNFR1 is strikingly impaired in ADAM17[3x/3x] hepatocytes upon PMA stimulation.**
Primary hepatocytes were stimulated with PMA (200 ng/ml) for 2 h in the presence or absence of MM (10 μM) and TNFR1 release was quantified via ELISA. * indicates a significant increase compared with control. # indicates a significant reduction in comparison with the stimulated sample. § indicates a significant reduction compared with the control group of other genotypes (/#/§: $P < 0.05$; n = 7; ±SEM). Data were analysed by two-way ANOVA and Bonferroni multiple comparison post hoc test.

ADAM17 KO mice ([18]). However, the early embryonic lethality precluded detailed phenotypic analysis of the other classical ADAM17 deficiency hallmarks. Thus, we cannot exclude that other factors might have contributed to the lethal phenotype. Early lethality was actually in accord with the observation that the time of death is influenced by the background of the mouse strains in ADAM17-deficient animals. Whereas mixed or pure 129 background results in mice reaching adulthood, the proportion of Bl6 background markedly shortens the life span ([14], [34]). At the commencement of this study, it seemed to us most unlikely that the consequences of mutating just 3 amino acids in the membrane-proximal domain would be nearly as profound as the deletion of the entire catalytic domain. The pure B16 background was, therefore, chosen to maximize the chances of detecting a more discrete phenotype. The unexpected findings made were remarkable because they indicate that ADAM17 sheddase activity truly depends on the presence of the small membrane-proximal PS-binding motif. On the basis of in vitro data, it is our bias that this is because binding of the motif to surface-exposed PS is essential for the protease to fulfil its sheddase function. However, we cannot totally exclude that mutation of the 3 amino acids affects interaction with other proteins such as iRhom2. Because of the lack of reliable antibodies against murine iRhom2, this aspect cannot be addressed at present.

It is known that small quantities of ADAM17 are released and present in soluble form in the circulation ([35], [36], [37], [38]). The question whether the soluble enzyme might subserve vital functions by cleaving substrates on other cells is controversially debated ([4], [36], [39], [40]). The ADAM17-3x mutant, although incapable of shedding membrane-anchored substrates, was shown to be enzymatically active and still capable of cleaving a peptide substrate that was present in the medium ([17]). The lethal phenotype of the ADAM17[3x/3x] mice now clearly indicates that any soluble ADAM17 that may be present cannot fulfil vitally important functions during embryogenesis.

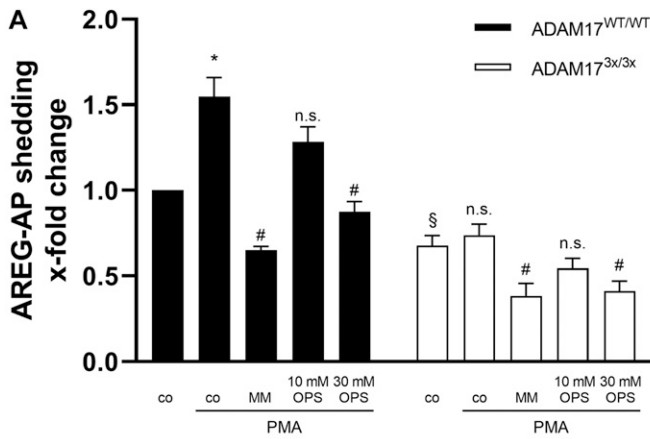

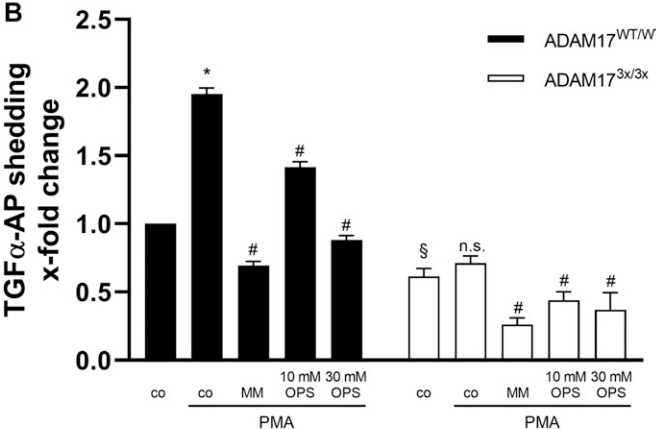

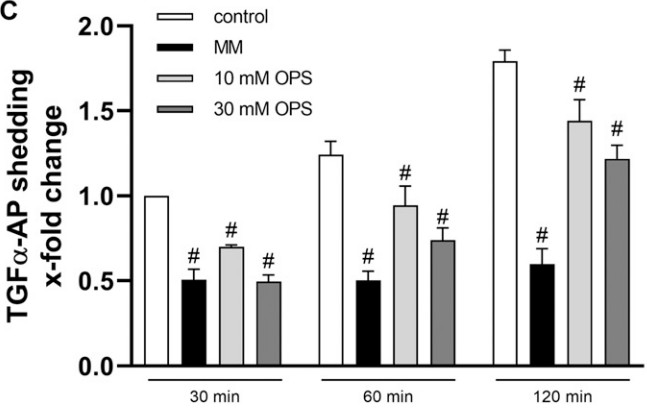

**Figure 4. PMA-stimulated release of ADAM17 substrates is abolished in ADAM17[3x/3x] MEFs.**
**(A, B)** MEFs were transfected with the AP-tagged ADAM17 substrates (A) AREG or (B) TGF-α, respectively. Cells were stimulated with PMA (200 ng/ml) for 120 min. PMA significantly increased the shedding in wild-type cells but not in ADAM17[3x/3x] cells. Broad-spectrum metalloprotease inhibitor MM (10 μM) and the PS headgroup OPS reduced the induced shedding in wild-type cells. **(C)** MEFs were transfected with TGF-α-AP and analysed for substrate release for the indicated time points in the presence of MM and OPS. **(A, B)** *indicates a significant increase compared with unstimulated cells, # indicates significant decrease compared with corresponding stimulated cells. § indicates a significant decrease of unstimulated ADAM17[3x/3x] compared with unstimulated wild-type cells. **(C)** # indicates significant decrease compared with corresponding control cells. **(A, B, C)**, (*/#/§: $P < 0.05$; [A, B, C] n = 4; [D] n = 3; ±SEM). Data were analysed by two-way ANOVA and Bonferroni multiple comparison post hoc test. n.s., no significant difference.

Our present data indicate that interaction of ADAM17 with PS is also important for constitutive substrate release. Thus, the release of soluble TNFR1 was absent in ADAM17$^{3x/3x}$ hepatocytes and fibroblasts. This finding was corroborated by the observation that release of both AREG and TGF-$\alpha$ was significantly lower in ADAM17$^{3x/3x}$ MEFs compared with WT cells. Moreover, OPS dose-dependently inhibited the constitutive release of TGF-$\alpha$ in wild-type fibroblasts. This complemented our previous data that had demonstrated massive increase of constitutive substrate release simply through enhancement of surface PS exposure ([20]).

Two facts merit emphasis in this context. First, PS is not exclusively localised in the inner cell membrane leaflet. Varying amounts of up to 20% total membrane PS can be found in the external leaflet depending on the cell type and status of differentiation in the absence of any stimuli ([41], [42], [43], [44], [45], [46], [47]). Second, PS exposure as one hallmark of apoptosis has greatly overshadowed the function of physiologic PS externalisation ([46]). Non-apoptotic PS exposure has been widely observed, for example, during sperm capacitation, myotube formation, macrophage-mediated phagocytosis, and activation of T lymphocytes, B lymphocytes, mast cells, and neutrophils ([48]). Moreover, several physiologic stimuli, including growth factors or ATP induce transient PS exposure within seconds to minutes ([17], [46], [49]). Thus, a certain amount of the phospholipid continuously traffics between the two membrane leaflets. Our data lead us to postulate that these small amounts suffice to contribute to the constitutive shedding activity of ADAM17. Further studies are required to address this aspect in detail.

Taken together, our results indicate that ADAM17 requires its PS-binding motif to fulfil its sheddase function. The data assign a biological role to translocated PS in controlling the action of membrane-anchored ADAM17, analogous to and as a counterpart of its major function as interaction partner for enzymes operative at the cytosolic membrane face.

# Materials and Methods

### Reagents and antibodies

Annexin V-Alexa 568 and Hoechst 33342 were purchased from Thermo Fisher Scientific. OPS and Phorbol 12-myristate-13-acetate (PMA) were obtained from Sigma-Aldrich. MM was purchased from Tocris Bioscience. Anti-ADAM17 antibodies used for Western blotting were from Abcam (ab57484) and Merck Millipore (AB19027). Additional antibodies used were tubulin (clone E7, DSHB), AKT (serine/threonine kinase 1/protein kinase B; Cell Signaling Technology), and EGFR (Cell Signaling Technology). Peroxidase-coupled secondary antibodies for immunoblot were from Jackson Immunoresearch. Anti-actin was from Santa Cruz. Antibodies used for FACS analyses were anti-ADAM17 from Abcam and secondary anti-mouse Alexa Fluor 488 from Invitrogen.

### Generation of ADAM17$^{3x/3x}$ mice

Mice were kept under specific pathogen-free conditions in isolated ventilated cages on a 12-h light/dark cycle with food and water ad libitum. Our investigations were carried out in accordance with the Guide for the Care and Use of Laboratory Animals of the German Animal Welfare Act on protection of animals. All animal protocols were approved by the relevant German authorities (project agreement no #997).

The constitutive knock-in point mutations R625G, K626G, and K628G were introduced into *ADAM17* exon 15 (National Center for Biotechnology Information transcript NM_009615.6) using CRISPR/Cas9-mediated gene editing with specific single guide RNA and oligonucleotide by Taconic. The Cas9 mRNA/protein, sgRNA and oligonucleotide were injected into C57BL/6NTac zygotes. An additional silent mutation (P629) was inserted into exon 15 to generate a restriction site (PpuMI) for analytical purposes. The correct gene editing was confirmed by sequencing of the target region. ADAM17$^{3x/3x}$ mice were maintained on a C57BL/6/NTac background. Wild-type littermates were used as controls. On the basis of the GRCm38/mm10 assembly (2011), off-target analysis has been performed. Potential non-intergenic off-target sites have been analysed in the G1 generation and could be excluded.

### Genotyping of ADAM17$^{3x/3x}$ mice

Genotyping was accomplished by isolating genomic DNA from tail tips, using DirectPCR Lysis Reagent (Peqlab) and following PCR and DNA digestion by restriction enzyme PpuMI (New England Biolabs).

The following primers were used:

5′-TCTGGGAGTAAGGCCAAAGAC-3′ (forward) and
5′-TGAGCTCAAAGACAGCAGACC-3′ (reverse).

PCR was performed for 35 cycles, 95°C for denaturation, 60°C for annealing, and 72°C for elongation performed with DNA Taq Polymerase (New England Biolabs) and a PeqStar Thermocycler (Peqlab). An aliquot of the PCR reaction was used to validate the presence or absence of the constitutive knock-in. DNA was digested using PpuMI. 10 U per enzyme were pipetted into the PCR reaction aliquot and incubated at the appropriate temperature for 48 h. A control plasmid demonstrated that the digest worked successfully.

The 543-bp PCR product was unaffected in wild types, partially digested in heterozygous mice, and completely digested in ADAM17$^{3x/3x}$ mice into two fragments (363 and 180 bp) as detected on an agarose gel.

### Primary cell culture

Primary hepatocytes were isolated from mice embryos at day E14. The embryo was washed in sterile PBS. The liver was separated from the rest of the embryonal material and washed in PBS. Afterwards, the embryonal liver cells were separated via up and down pipetting in 200 $\mu$l of DMEM (Thermo Fisher Scientific) with 10% FCS and 1% penicillin/streptomycin and transferred onto a collagen-coated cell culture dish. After 12 h, cell debris and unattached cells were removed and primary hepatocytes gently washed with PBS. Analysis of the cells was performed after 72 h of cultivation.

Primary MEFs were derived from dissociated E14 embryos and maintained in DMEM supplemented with FCS and antibiotics. Afterwards, the remaining tissue of the embryo was washed in PBS and minced into small pieces with an scalpel on a cell culture dish. The cultivation medium was DMEM with 10% FCS and

1% penicillin/streptomycin. After 12 h, the cell debris and unattached cells were removed and analysis was performed after 72 h.

## Biotinylation

For detection of cell surface–expressed proteins, isolation via Pierce Cell Surface Protein Isolation Kit (Thermo Fisher Scientific) was performed according to the manufacturer's instructions with two additional washing steps using Gibco Dulbecco's Phosphate Buffered Saline (Thermo Fisher Scientific) after quenching the biotinylation reaction. Analysis of the cell surface fraction was performed via immunoblot.

## ELISA

TNFR1 ELISA (R&D) was performed according to the manufacturer's instructions. Primary hepatocytes were directly seeded into 12-well plates and grown until confluence. Hepatocytes were stimulated with PMA (200 ng/ml) in the presence or absence of MM (10 $\mu$M) for 2 h. Supernatants were analysed for soluble shedding products and cell lysates for full-length product in duplicates. Shown is the relative amount of TNFR1 in the supernatant compared with the total amount of supernatant plus cell lysates. Defined numbers of MEFs were seeded into six-well plates and stimulated with PMA (200 ng/ml) in the presence or absence of MM (10 $\mu$M) for 2 h after 24 h of culture. The analysis of cell lysates ensured comparable expression of TNFR1 in all genotypes (not shown). Supernatants were analysed for soluble shedding products in duplicates.

## Western blot analysis

Cells were washed once with PBS and lysed in lysis buffer (5 mM Tris–HCl (pH 7.5), 1 mM EGTA, 250 mM saccharose, and 1% Triton X-100) supplemented with cOmplete Inhibitor Cocktail (Roche Applied Science) and 10 mM 1,10-phenanthroline monohydrate to prevent ADAM17 autocleavage ([22]). In addition, sonification of cell lysates was performed. Equal amounts of protein were loaded on 10% SDS–PAGE gels. The samples were electrotransferred onto polyvinylidene difluoride membranes (Hybond-P; Amersham) and blocked overnight with 5% skim milk in TBS. After incubation with the indicated antibody in blocking buffer, the membranes were washed three times in TBST (TBS containing 0.1% Tween-20). Primary antibodies were detected using affinity-purified peroxidase-conjugated secondary antibodies (1:10.000) for 1 h at room temperature. Detection was carried out using the ECL detection system (Amersham). Signals were recorded by a luminescent image analyser (Fusion FX7 imaging system; PEQLAB Biotechnologie). To analyse the expression of different antigens on the same polyvinylidene difluoride membrane, Western blots were incubated in stripping reagent (100 mM 2-mercaptoethanol, 2% [wt/vol] SDS, 62.5 mM, and Tris–HCl, pH 6.7) at 55°C for 30 min and re-probed.

## Flow cytometric analysis

For analysing ADAM17 cell surface expression, hepatocytes were stained with an anti-ADAM17 antibody (57484; Abcam) (1:50) or isotype control antibody (mouse IgG2b; R&D MAB004) in PBS with

10% normal goat serum (Sigma-Aldrich) for 60 min on ice. After washing, the cells were incubated with antimouse Alexa Fluor 488 (1:250) for 30 min on ice. After staining, the cells were washed and analysed by flow cytometry with FACSVerse (BD Bioscience). Data were analysed using FlowJo software (Version 10.6.1).

## Annexin staining

Primary hepatocytes and MEFs were seeded on $\mu$-slide ibidi glass chambers and grown to semi-confluence. After indicated stimulation periods, each chamber was immediately incubated with a 1:20 solution of Annexin V-568 in annexin binding buffer for 5 min in the dark at room temperature, washed twice with annexin binding buffer (10 mM Hepes, 140 mM NaCl, and 2.5 mM CaCl$_2$, pH 7.4) and fixed for 30 min with 3% PFA. After fixation, the slides were washed 3 times with PBS and Hoechst 33342 (1:5.000 in PBS) was incubated for 10 min. Afterwards, the fixated cells were washed 3 times with PBS, once with distilled water, and covered with mounting medium (ibidi). Image acquisition was performed with an inverted confocal microscope (Fluoview FV1000, Olympus) using a UPLSAPO 60× oil immersion objective (NA:1.35) and 2.5× zoom. Images were acquired with the same laser and detection settings for each experimental setup.

## Transfection and AP substrate shedding assay

MEFs were transfected using Lipofectamine 3000 (Thermo Fisher Scientific) according to the manufacturer's instructions. 48 h after transfection, the cells were treated or not treated with PMA (200 ng/ml, 120 min) in the absence or presence of MM (10 $\mu$M) or OPS (10 or 30 mM). Thereafter, supernatants and cell lysates were collected and measured for AP activity at A405, using the AP substrate 4-nitrophenyl phosphate (Sigma-Aldrich). For constitutive shedding analyses, the cells were washed 48 h after transfection and incubated in DMEM for the indicated time points in the absence or presence of MM (10 $\mu$M) or OPS (10 or 30 mM). Thereafter, supernatants and cell lysates were analysed for AP activity. Shown is the relative AP activity in the supernatant compared with the total AP activity of supernatant plus cell lysates.

## Image analysis and image statistics

Image analysis was performed with ImageJ version 1.51. Fluorescence signal above background fluorescence was determined and correlated with the cell number. Three different animals per genotype were either treated with PMA or untreated and analysed for their fluorescent signal. The mean fluorescence per cell number was taken for statistical analysis. Groups were tested by two-way ANOVA and Bonferroni multiple comparison post hoc test.

## Statistical analysis

All values for the ectodomain shedding assay and the fluorescent quantification are expressed as means ± SEM. The standard error values indicate the variation between mean values obtained from at least three independent experiments/animals per genotype. Statistics were generated using two-way ANOVA and Bonferroni

multiple comparison post hoc test. *P* values < 0.05 were considered statistically significant (either indicated with *, # or §).

# Supplementary Information

# Acknowledgements

This work was funded by the Deutsche Forschungsgemeinschaft (German Research Foundation)—project number 125440785—SFB877 (A4; K Reiss and M Veit) and supported by RTG1743 (J Seidel). The microscope platform was kindly provided by the SFB877 project Z3. We thank the Institute of Clinical Molecular Biology in Kiel for providing Sanger sequencing as supported in part by the Cluster of Excellence "Inflammation at Interfaces" and "Future Ocean". We thank the Animal Facility for housing and breeding of our mice.

## Author Contributions

M Veit: data curation, investigation, visualization, and methodology.
B Ahrens: investigation.
J Seidel: investigation.
A Sommer: conceptualization.
S Bhakdi: conceptualization and writing—original draft.
K Reiss: conceptualization, supervision, funding acquisition, and writing—original draft.

## Conflict of Interest Statement

The authors declare that they have no conflict of interest.

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
