## [Reviewer comments · Life Science Alliance]

Life Science Alliance

Mutagenesis of the ADAM17-phosphatidylserine-binding motif leads to embryonic lethality in mice

Martin Veit, Björn Ahrens, Jana Seidel, Anselm Sommer, Sucharit Bhakdi, and Karina Reiss

DOI: <https://doi.org/10.26508/lsa.201900430>

Corresponding author(s): Karina Reiss, University Hospital Schleswig-Holstein. Campus Kiel

Review Timeline:	Submission Date:	2019-05-15
	Editorial Decision:	2019-06-07
	Revision Received:	2019-08-08
	Editorial Decision:	2019-08-16
	Revision Received:	2019-08-19
	Accepted:	2019-08-19

Scientific Editor: Andrea Leibfried

Transaction Report:

June 7, 2019

Re: Life Science Alliance manuscript #LSA-2019-00430-T

Prof. Karina Reiss
University Hospital Schleswig-Holstein. Campus Kiel
Department of Dermatology and Allergology
Schittenhelmstrasse 7
Kiel D-24105
Germany

Dear Dr. Reiss,

Thank you for submitting your manuscript entitled "Mutagenesis of the ADAM17-phosphatidylserine-binding motif leads to embryonic lethality in mice" to Life Science Alliance. The manuscript was assessed by expert reviewers, whose comments are appended to this letter.

As you will see, the reviewers appreciate the mouse model generated and the analyses performed, and they provide constructive input on how to further strengthen your study to allow publication here. I would thus like to invite you to provide a revised manuscript, addressing the individual concerns raised. These are partially overlapping and seem straightforward to address. Importantly, the requested text changes and further discussion need to get introduced, the shedding of endogenous / more relevant ADAM17 substrates needs to get tested, and the MEFs need to get further analyzed and controls performed.

Reviewer #3 provided a pdf of your manuscript highlighting spelling errors, which can be found in lines:

35 (punctuation)
81 (wrong figure mentioned)
87 (genotype description)
122 (wrong figure mentioned)
126 (wrong figure mentioned)
137 (references; wrong figure mentioned)
180 (superfluous space before degree signs)
184 (missing space before 'U')

Thank you for this interesting contribution to Life Science Alliance. We are looking forward to receiving your revised manuscript.

Sincerely,

B. MANUSCRIPT ORGANIZATION AND FORMATTING:

We encourage our authors to provide original source data, particularly uncropped/-processed electrophoretic blots and spreadsheets for the main figures of the manuscript. If you would like to add source data, we would welcome one PDF/Excel-file per figure for this information. These files

will be linked online as supplementary "Source Data" files.

Reviewer #1 (Comments to the Authors (Required)):

The cell surface protease ADAM17 catalyzes the release of important signaling molecules, including the inflammatory cytokine TNF and the activating ligands of the epidermal growth factor receptor (EGFR). A substantial body of literature has been devoted to understanding the molecular basis for how the activity of ADAM17 is stimulated. In spite of this, almost none of these findings have been addressed in an organismal context. Hence, the physiological basis of ADAM17 regulation in vivo remains unclear.

Here, Veit and colleagues focus on the hypothesis that a phosphatidylserine (PS) binding motif within ADAM17 is essential for the physiological stimulation of its activity in vivo. This follows on from a cellular study from the same group (Sommer et al., Nat. Communications, 2016) that demonstrated that the membrane proximal domain of ADAM17 has PS binding sites. PS is normally restricted asymmetrically to the inner leaflet of the bilayer and is not found on the outer leaflet. The model proposed by Sommer et al. was that stimuli that promote ADAM17 activity trigger localized PS flipping to the outer leaflet. Externalized PS recruits ADAM17 in close proximity to its substrates, enabling efficient stimulated substrate cleavage. Hence, PS exposure and ADAM17 recruitment to the membrane were proposed as critical step prior to proteolysis in response to ADAM17 stimulants.

The present MS follows up on these cellular experiments by generating/characterizing a mouse mutant in which the PS-binding motifs in ADAM17 are mutated, to abrogate PS binding. This approach is an important addition to the field, for the reasons noted above. Interestingly, these mutant animals exhibit embryonic lethality, presenting with brain hemorrhages (a phenotype described for some ADAM17 KO models) at E14, somewhat earlier than expected compared to other models of ADAM17 deficiency.

Westerns from tissues from these mutant animals show that the maturation and the cell surface localization of ADAM17 is normal, suggesting that the trafficking of mutant ADAM17 within the secretory pathway is also normal. Importantly, the authors confirm that the proteolytic activity of ADAM17 in primary hepatocytes is defective in homozygous cells from the mutant, in spite of the cells being capable of externalizing PS as efficiently as WT cells.

Overall, this is an interesting and potentially important study, because as noted above, the mechanism for ADAM17 has been studied almost exclusively in vitro to date. Although the data presented here (noted above) seem robust and consistent with defective ADAM17 shedding, there are a number of anomalies that the authors should address more openly.

1. The tone of some of the text is unduly dogmatic. For example, p5 "they leave virtually no room for doubt"; "we submit that the black or white, all or nothing response to ADAM17 expression renders interpretation of the present data quite unequivocal"; p7 "PMA is the only stimulus that

activates ADAM17 and not the major related protease, ADAM10". This tone should be corrected since the mutant dies earlier than anticipated from other ADAM17 studies, at E14 and no detailed phenotypic data is provided that demonstrates unambiguously an hallmark ADAM17 phenotype. As it is not possible to assay for the classical ADAM17 phenotypes (e.g. eyes open at birth, lung morphogenesis defects, hair follicle defects) that manifest later during development, the paper does not reach the burden of proof to entitle the use of such dogmatic language. While the animals indeed exhibit brain haemorrhages which have been observed in some ADAM17 or EGFR mutant mouse models, many other genetic mutations and pathological conditions also give rise to brain haemorrhages. Hence, some caution in the narrative is warranted.

2. As the brain phenotype is the only potential ADAM17-related phenotype studied, the authors should indicate how penetrant the phenotype is.

3. Why is the constitutive activity of ADAM17 also affected, and not just the stimulated activity? (Fig. 3) We know that PS is not normally found on the outer leaflet in unstimulated conditions. Why then is basal shedding affected? This does not fit the model, and is an important consideration that should be discussed.

4. In the absence of a frank organismal ADAM17-related phenotype, which the genetic background perhaps excludes, it would be useful to determine whether more than one substrate is affected in ADAM17 3x primary cells. The authors have derived MEFs, so the shedding of endogenous or model ADAM17 substrates could be studied in this setting. This should be do-able within 1-2 months, assuming that MEFs are available and/or animals are available to generate the MEFs.

Reviewer #2 (Comments to the Authors (Required)):

The protease ADAM17 has a key role in inflammation and tissue homeostasis and is a major drug target for different conditions. It is known to be rapidly activatable, but the underlying mechanisms appear diverse and are not yet well understood. This paper is a follow-up of previous work from the same group, where the authors demonstrated a new mechanism in vitro for ADAM17 activation. Now, Veit et al. provide in vivo evidence for this mechanism. They report the phenotype of a novel ADAM17 mutant mouse model (ADAM17 3x/3x) which harbors three point mutations in a putative phosphatidylserine (PS)-binding motif (R625G, K626G and K628G). The mice reveal a severe developmental phenotype and die between embryonic days E14-15. ADAM17 mutant mice which lack the catalytic site reveal a similarly severe phenotype depending on their genetic background. Therefore, Veit et al. suggest their ADAM17 3x/3x resemble ADAM17 loss of function mutant mice. The ADAM17 3x/3x mouse model is novel, exciting and provides an excellent new tool to gain new insights into the ADAM17 activation mechanism in vivo. The phenotype of the mice is very striking and unexpectedly strong. Yet, the conclusions are not yet justified by the data. The authors make very bold statements (e.g. The results "leave virtually no room for doubt....."; or "we provide in vivo evidence for the essential role of PS-interaction.....") at several occasions throughout the manuscript, which overinterpret the data. For example, the manuscript does not provide any in vivo evidence for the essential role of PS-interaction in vivo. All that the authors demonstrate is that the lack of a sequence motif (required for PS binding in vitro) appears to abolish ADAM17 activity. Thus, it is clear that this motif is essential for activity. But the authors do not provide any proof for the involvement of PS in vivo. Moreover, they completely disregard other key findings in the field, which may provide alternative mechanistic explanations. For example, the mutation in ADAM17 may affect levels or interaction with iRhom2 or FRMD8, essential ADAM17-associated proteins.

The following points require attention for improving the manuscript.

Major Points:

1. The manuscript should be reorganized and have distinct sections for Results and Discussion. This would allow the authors to first describe their beautiful and exciting data without the need of overinterpretation. Subsequently, in the discussion section the authors may speculate about the underlying mechanisms. Here they can make the case for the relevance of the PS binding, but can easily consider other or additional mechanisms as well. But, even if they do not find further evidence to proof the role of PS binding in vivo, the data presented are spectacular and do not require overinterpretation.
2. The authors need to rule out alternative mechanistic scenarios other than PS. Thus, they need to blot for iRhom2 (for example in lymphocytes or macrophages) and demonstrate that iRhom2 levels are unaltered.
3. The authors fail to show a direct link between the 3x mutations and PS in vivo. More evidence for this interaction in vivo is required. Otherwise, the paper needs to be rephrased and the part on PS needs to be toned down. Again, I would like to emphasize that the reported findings by themselves are exciting.
4. Figure 2 A shows Western blots of mature and proADAM17 in different cell types, including at the cell surface. These experiments are essential for their claim that the mutation does not simply alter trafficking of ADAM17, thus preventing it from reaching the cell surface. The authors conclude that levels of ADAM17 are not altered due to the mutation. While this conclusion may be justified for liver, brain and hepatocytes, it is not justified for MEFs. In MEFs, the authors demonstrate three bands - an upper one and two lower ones. The upper one of the two lower ones (a bit fainter and smeary) is absent in ADAM17 3x/3x MEFs. But this typically constitutes the mature ADAM17 band. Thus, my conclusion would be that in MEFs there is no mature mutant ADAM17. This, would indicate, however, that the mutation prevents ADAM17 maturation and thus, activity, and could be independent of the PS-based mechanism. To provide evidence for the specificity of the detected bands, the authors need to provide samples from liver, brain, MEFs and hepatocytes of ADAM17 KO mice, which they have available. Otherwise, I am afraid that the wrong band may have been used for interpreting the data. In an additional experiment the authors may even consider to use a lysis buffer without phenantroline, which may also serve as a negative control for the mature band of ADAM17, as the lack of phenanthroline allows autocatalytic degradation of ADAM17.
5. Figure 2 B: The biotinylation experiment is not convincing and needs to be repeated with another cell type like MEFs. In MEFs it is known that only the mature band of ADAM17 is detectable on the cell surface. Thus, to rule out a cell-specific effect just in hepatocytes, the MEF data are required. In addition, since it is a key statement of the paper, the maturation of ADAM17^{3x/3x} to the cell surface should be proven by an alternative method such as FACS analysis, which has been used by other groups before.

Minor points:

1. The authors discuss and compare their phenotype only to the ADAM17 mutant mouse, which lacks the catalytic site. However, other ADAM17 loss of function mutant mouse models have been described and should be added to the comparison/discussion (e.g. ADAM17 KO, ADAM17 ex/ex, iRhom1 & 2 DKO....).
2. Refer in the text to the accurate Figure number (lines 122, 81)
3. Enlarge the area of the Western Blot pictures to be shown in the final figure so that the reader may more easily be convinced of the true identity of the bands.
4. Line 132: delete the word ONLY. There are other stimuli as well that only act on ADAM17 not on ADAM10.

Reviewer #3 (Comments to the Authors (Required)):

Summary:

The manuscript by Veit et al describes the influence of the ADAM17-phosphatidylserine-binding motif in newly generated knock-in mice. Based on their previous findings, the authors generated a new knock-in mice for an ADAM17 variant which lacks the PS binding motif. ADAM17-deltaPS has been shown to be catalytically inactive. Therefore it might come without a big surprise that this knock-in mice more or less reflects previous findings, e.g. embryonic lethality. The authors convincingly showed that expression of ADAM17deltaPS was comparable to expression of wild-type ADAM17 (pro- and mature form). They also showed that cells from these mice were not able to shed TNFR1.

General remark:

This is a very straightforward study. The manuscript is well written and the conclusions drawn from the experiments are sound. One might argue that some more characterization of the overall phenotype of the mice could have been done in more detail, this can hardly be done at E14 and might give not too much additional information. Albeit the authors show shedding of TNFR1 this is not the most critical substrate which causes this phenotype. Therefore shedding of more relevant target protein would be desirable.

Minor comments:

1. I have marked typos directly in the pdf of the manuscript (red), which could be obtained upon request. It was not possible to upload any documents.
2. Statistical significance should be calculated for Figure 3 between the control conditions of all genotypes.
3. The Figures and some references on page 7 have been mixed up.

Reviewer #1

The cell surface protease ADAM17 catalyzes the release of important signaling molecules, including the inflammatory cytokine TNF and the activating ligands of the epidermal growth factor receptor (EGFR). A substantial body of literature has been devoted to understanding the molecular basis for how the activity of ADAM17 is stimulated. In spite of this, almost none of these findings have been addressed in an organismal context. Hence, the physiological basis of ADAM17 regulation in vivo remains unclear.

Here, Veit and colleagues focus on the hypothesis that a phosphatidylserine (PS) binding motif within ADAM17 is essential for the physiological stimulation of its activity in vivo. This follows on from a cellular study from the same group (Sommer et al., Nat. Communications, 2016) that demonstrated that the membrane proximal domain of ADAM17 has PS binding sites. PS is normally restricted asymmetrically to the inner leaflet of the bilayer and is not found on the outer leaflet. The model proposed by Sommer et al. was that stimuli that promote ADAM17 activity trigger localized PS flipping to the outer leaflet. Externalized PS recruits ADAM17 in close proximity to its substrates, enabling efficient stimulated substrate cleavage. Hence, PS exposure and ADAM17 recruitment to the membrane were proposed as critical step prior to proteolysis in response to ADAM17 stimulants. The present MS follows up on these cellular experiments by generating/characterizing a mouse mutant in which the PS-binding motifs in ADAM17 are mutated, to abrogate PS binding. This approach is an important addition to the field, for the reasons noted above. Interestingly, these mutant animals exhibit embryonic lethality, presenting with brain hemorrhages (a phenotype described for some ADAM17 KO models) at E14, somewhat earlier than expected compared to other models of ADAM17 deficiency. Westerns from tissues from these mutant animals show that the maturation and the cell surface localization of ADAM17 is normal, suggesting that the trafficking of mutant ADAM17 within the secretory pathway is also normal. Importantly, the authors confirm that the proteolytic activity of ADAM17 in primary hepatocytes is defective in homozygous cells from the mutant, in spite of the cells being capable of externalizing PS as efficiently as WT cells. Overall, this is an interesting and potentially important study, because as noted above, the mechanism for ADAM17 has been studied almost exclusively in vitro to date. Although the data presented here (noted above) seem robust and consistent with defective ADAM17 shedding, there are a number of anomalies that the authors should address more openly.

RESPONSE: We thank the reviewer for wonderfully summarizing the state of the art.

1. The tone of the some of the text is unduly dogmatic. For example, p5 "they leave virtually no room for doubt"; "we submit that the black or white, all or nothing response to ADAM17 expression renders interpretation of the present data quite unequivocal"; p7 "PMA is the only stimulus that activates ADAM17 and not the major related protease, ADAM10". This tone should be corrected since the mutant dies earlier than anticipated from other ADAM17 studies, at E14 and no detailed phenotypic data is provided that demonstrates unambiguously an hallmark ADAM17 phenotype. As it is not possible to assay for the classical ADAM17 phenotypes (e.g. eyes open at birth, lung morphogenesis defects, hair follicle defects) that manifest later during development, the paper does not reach the burden of proof to entitle the use of such dogmatic language. While the animals indeed exhibit brain haemorrhages which have been observed in some ADAM17 or EGFR mutant mouse models, many other genetic mutations and pathological conditions also give rise to brain haemorrhages. Hence, some caution in the narrative is warranted.

RESPONSE: The reviewer is right. Since a detailed phenotypic characterisation is not possible due to the early embryonic death of the mice, we have toned down our statements and alluded to this shortcoming in the Discussion.

2. As the brain phenotype is the only potential ADAM17-related phenotype studied, the authors should indicate how penetrant the phenotype is.

RESPONSE: The penetrance of the brain phenotype is about 50%, comparable with the classical knockout. This information has been added in Results.

3. Why is the constitutive activity of ADAM17 also affected, and not just the stimulated activity? (Fig. 3) We know that PS is not normally found on the outer leaflet in unstimulated conditions. Why then is basal shedding affected? This does not fit the model, and is an important consideration that should be discussed.

RESPONSE: We thank the reviewer for pointing out this interesting finding. Additional experiments (new Fig 4A-C and Supplementary Fig S4C) are now presented. They indeed indicate that ADAM17-PS interaction is also involved in constitutive shedding. The possible explanation for this finding is discussed (page 7) as follows:

“Our present data indicate that interaction of ADAM17 with PS is also important for constitutive substrate release. Thus, the release of soluble TNFR1 was absent in ADAM17-3x/3x hepatocytes and fibroblasts. This finding was corroborated by the observation that release of both AREG and TGF-alpha was significantly lower in ADAM17-3x/3x MEFs compared to WT cells. Moreover, OPS dose-dependently inhibited the constitutive release of TGF-alpha in wildtype fibroblasts. This complemented our previous data that had demonstrated massive increase of constitutive substrate release simply through enhancement of surface PS exposure [20].

Two facts merit emphasis in this context. First, PS is not exclusively localized in the inner cell membrane leaflet. Varying amounts of up to 20% total membrane PS can be found in the external leaflet depending e.g. on the cell type and status of differentiation in the absence of any stimuli [41-47]. Second, PS exposure as one hallmark of apoptosis has greatly overshadowed the function of physiologic PS externalisation [46]. Non-apoptotic PS exposure has been widely observed, e.g. during sperm capacitation, myotube formation, macrophage-mediated phagocytosis and during activation of T lymphocytes, B lymphocytes, mast cells and neutrophils [48]. Moreover, several physiologic stimuli including growth factors or ATP induce transient PS-exposure within seconds to minutes [17,46,49]. Thus, a certain amount of the phospholipid continuously traffics between the two membrane leaflets. Our data lead us to postulate that these small amounts suffice to contribute to the constitutive shedding activity of ADAM17. Further studies are required to address this aspect in detail.”

4. In the absence of a frank organismal ADAM17-related phenotype, which the genetic background perhaps excludes, it would be useful to determine whether more than one substrate is affected in ADAM17 3x primary cells. The authors have derived MEFs, so the shedding of endogenous or model ADAM17 substrates could be studied in this setting. This should be do-able within 1-2 months, assuming that MEFs are available and/or animals are available to generate the MEFs.

RESPONSE: As proposed by the reviewer, we additionally performed ELISAs for endogenous TNFR1 in MEFs (new Supplementary Figure S4) and we used these cells for AP-Assays with TGF-alpha and AREG (new Fig 4). All the data are in line with the previous findings.

Reviewer #2

1. *The manuscript should be reorganized and have distinct sections for Results and Discussion. This would allow the authors to first describe their beautiful and exciting data without the need of overinterpretation. Subsequently, in the discussion section the authors may speculate about the underlying mechanisms. Here they can make the case for the relevance of the PS binding, but can easily consider other or additional mechanisms as well. But, even if they do not find further evidence to proof the role of PS binding in vivo, the data presented are spectacular and do not require overinterpretation.*

Response: We thank the reviewer for the positive feedback. As suggested, we reorganized the manuscript and tried to avoid any over-interpretation. In addition, we extended our results and show additional experiments underlining the relevance of PS interaction for ADAM17-mediated substrate release (new Fig 4 and Supplementary Fig S4).

2. *The authors need to rule out alternative mechanistic scenarios other than PS. Thus, they need to blot for iRhom2 (for example in lymphocytes or macrophages) and demonstrate that iRhom2 levels are unaltered.*

Response: Unfortunately, the early lethality of the embryos at E14 makes it impossible to prepare lymphocytes or macrophages. In order to address the reviewers concern, we thus planned to analyse iRhom2 protein expression in hepatocytes and MEFs. However, we then learned from experts in the iRhom field that none of the commercially available antibodies can be considered trustworthy because all generate the same staining pattern in iRhom2 knockout cells.

3. *The authors fail to show a direct link between the 3x mutations and PS in vivo. More evidence for this interaction in vivo is required. Otherwise, the paper needs to be rephrased and the part on PS needs to be toned down. Again, I would like to emphasize that the reported findings by themselves are exciting.*

Response: We toned down our statements and rephrased the Discussion as requested.

4. *Figure 2 A shows Western blots of mature and proADAM17 in different cell types, including at the cell surface. These experiments are essential for their claim that the mutation does not simply alter trafficking of ADAM17, thus preventing it from reaching the cell surface. The authors conclude that levels of ADAM17 are not altered due to the mutation. While this conclusion may be justified for liver, brain and hepatocytes, it is not justified for MEFs. In MEFs, the authors demonstrate three bands - an upper one and two lower ones. The upper one of the two lower ones (a bit fainter and smeary) is absent in ADAM17 3x/3x MEFs. But this typically constitutes the mature ADAM17 band. Thus, my conclusion would be that in MEFs there is no mature mutant ADAM17. This, would indicate, however, that the mutation prevents ADAM17 maturation and thus, activity, and could be independent of the PS-based mechanism. To provide evidence for the specificity of the detected bands, the authors need to provide samples from liver, brain, MEFs and hepatocytes of ADAM17 KO mice, which they have available. Otherwise, I am afraid that the wrong band may have been used for interpreting the data. In an additional experiment the authors may even consider to use a lysis buffer without phenantroline, which may also serve as a negative control for the mature band of ADAM17, as the lack of phenanthroline allows autocatalytic degradation of ADAM17.*

Response: We agree with the reviewer that the cell surface localisation of ADAM17 is of major importance. We therefore further characterized the hepatocytes using flow cytometric analyses. These confirmed that trafficking of ADAM17 was not perturbed in these cells (new Supplementary Figure S2). It did not seem very likely to us that trafficking should not be affected in hepatocytes but in MEFs. However, we also realized that the antibody led to a 3 band pattern in WT but not in ADAM17-3x fibroblasts. To clarify the situation we performed biotinylation experiments with our

MEFs. These showed that the upper band of the two lower ones is obviously something unspecific. More importantly, we found that the amounts of mature cell surface ADAM17 were comparable between the genotypes.

5. Figure 2 B: The biotinylation experiment is not convincing and needs to be repeated with another cell type like MEFs. In MEFs it is known that only the mature band of ADAM17 is detectable on the cell surface. Thus, to rule out a cell-specific effect just in hepatocytes, the MEF data are required. In addition, since it is a key statement of the paper, the maturation of ADAM17-3x/3x to the cell surface should be proven by an alternative method such as FACS analysis, which has been used by other groups before.

Response: See above.

Minor points:

1. The authors discuss and compare their phenotype only to the ADAM17 mutant mouse, which lacks the catalytic site. However, other ADAM17 loss of function mutant mouse models have been described and should be added to the comparison/discussion (e.g. ADAM17 KO, ADAM17 ex/ex, iRhom1 &2 DKO....).

Response: Other loss of function mutants are now alluded to as requested (page 7).

2. Refer in the text to the accurate Figure number (lines 122, 81)

Response: Our apologies. Correction has been undertaken.

3. Enlarge the area of the Western Blot pictures to be shown in the final figure so that the reader may more easily be convinced of the true identity of the bands.

Response: We enlarged the area of the Western Blot pictures. In addition with our new experiments (flow cytometry for hepatocytes and biotinylation of MEFs) the situation appears quite clear.

4. Line 132: delete the word ONLY. There are other stimuli as well that only act on ADAM17 not on ADAM10.

Response: Done.

Reviewer #3

General remark:

This is a very straightforward study. The manuscript is well written and the conclusions drawn from the experiments are sound. One might argue that some more characterization of the overall phenotype of the mice could have been done in more detail, this can hardly be done at E14 and might give not too much additional information. Albeit the authors show shedding of TNFR1 this is not the most critical substrate which causes this phenotype. Therefore shedding of more relevant target protein would be desirable.

Minor comments:

1. I have marked typos directly in the pdf of the manuscript (red), which could be obtained upon request. It was not possible to upload any documents.

Response: Thank you very much for your kind help.

2. Statistical significance should be calculated for Figure 3 between the control conditions of all genotypes.

Response: The statistical differences for the control conditions of all genotypes are now directly indicated in Figure 3 and Supplementary Figure S4C.

3. The Figures and some references on page 7 have been mixed up.

Response: Our apologies. The mistakes have been corrected.

August 16, 2019

RE: Life Science Alliance Manuscript #LSA-2019-00430-TR

Prof. Karina Reiss
University Hospital Schleswig-Holstein. Campus Kiel
Department of Dermatology and Allergology
Rosalind-Franklin-Straße 7
Kiel D-24105
Germany

Dear Dr. Reiss,

Thank you for submitting your revised manuscript entitled "Mutagenesis of the ADAM17-phosphatidylserine-binding motif leads to embryonic lethality in mice". As you will see, reviewer #2 appreciates the changes introduced in revision and we would thus be happy to accept your manuscript for publication in Life Science Alliance, pending final revisions to provide missing information (rev#2) and to match our guidelines:

- Please add the information for the control FACS staining in Fig. S2 (reviewer #2)
- Please add a section in the M&M to confirm that all animal experiments were performed in accordance with relevant guidelines and regulations and please include information on the institutional and/or licensing committee approving the experiments
- I think it would be good to add "in mice" to the following sentence in your abstract:
Here, we show that mutagenesis of the three amino acids constituting the PS-binding motif leads to embryonic lethality.

A. FINAL FILES:

-- High-resolution figure, supplementary figure and video files uploaded as individual files: See our detailed guidelines for preparing your production-ready images, <http://www.life-science->

alliance.org/authors

B. MANUSCRIPT ORGANIZATION AND FORMATTING:

Sincerely,

Andrea Leibfried, PhD
Executive Editor
Life Science Alliance
Meyershofstr. 1
69117 Heidelberg, Germany
t +49 6221 8891 502
e a.leibfried@life-science-alliance.org

Reviewer #2 (Comments to the Authors (Required)):

The authors have adequately addressed most of my previously raised points. Although they are still not providing Western blot band patterns from WT and ADAM17 KO MEFs, they did additional experiments (surface biotinylation) that appear to allow the appropriate conclusion. Yet, one experiment needs to be better described. For the FACS staining in Fig. S2 it is unclear to me what the control is. The authors describe this as mock cells, but since they did not use transfections, it remains unclear what mock means. Were they labeled only with the secondary antibody? And can they provide a citation that the ADAM17 antibody is specific in the FACS analysis? This should be added to the figure legend or methods section.

Reviewer #2

Reviewer #2 (Comments to the Authors (Required)):

The authors have adequately addressed most of my previously raised points. Although they are still not providing Western blot band patterns from WT and ADAM17 KO MEFs, they did additional experiments (surface biotinylation) that appear to allow the appropriate conclusion. Yet, one experiment needs to be better described. For the FACS staining in Fig. S2 it is unclear to me what the control is. The authors describe this as mock cells, but since they did not use transfections, it remains unclear what mock means. Were they labeled only with the secondary antibody? And can they provide a citation that the ADAM17 antibody is specific in the FACS analysis? This should be added to the figure legend or methods section.

Response: We agree that the description of the FACS analysis was not sufficiently precise. To make it clear, we now included the isotype controls for both genotypes and the control of unstained WT cells (only secondary antibody) and provide a more detailed description in the Figure legend. The abcam ADAM17 antibody has been used by other groups also for FACS analysis (e.g. Jiang et al. *J Immunol* 2017). However, everyone can find such information as well as background information for the antibody on the providers homepage. Thus, we do not think that such details should go into a figure legend or methods section.

August 19, 2019

RE: Life Science Alliance Manuscript #LSA-2019-00430-TRR

Prof. Karina Reiss
University Hospital Schleswig-Holstein. Campus Kiel
Department of Dermatology and Allergology
Rosalind-Franklin-Straße 7
Kiel D-24105
Germany

Dear Dr. Reiss,

Thank you for submitting your Research Article entitled "Mutagenesis of the ADAM17-phosphatidylserine-binding motif leads to embryonic lethality in mice". I appreciate the introduced changes and it is a pleasure to let you know that your manuscript is now accepted for publication in Life Science Alliance. Congratulations on this interesting work.

DISTRIBUTION OF MATERIALS:

Again, congratulations on a very nice paper. I hope you found the review process to be constructive and are pleased with how the manuscript was handled editorially. We look forward to future exciting submissions from your lab.

Sincerely,

Andrea Leibfried, PhD
Executive Editor
Life Science Alliance
Meyerohofstr. 1
69117 Heidelberg, Germany
t +49 6221 8891 502
e a.leibfried@life-science-alliance.org
www.life-science-alliance.org